# From Lithological Modelling to Groundwater Modelling: A Case Study in the Tiber River Alluvial Valley

**Cristina Di Salvo** [1,*], **Marco Mancini** [1,*], **Massimiliano Moscatelli** [1], **Maurizio Simionato** [1], **Gian Paolo Cavinato** [1], **Michele Dimasi** [2] and **Francesco Stigliano** [1]

1   CNR-Institute of Environmental Geology and Geoengineering, Area della Ricerca Roma 1, Via Salaria Km 29,300, Monterotondo Stazione, 00015 Rome, Italy; massimiliano.moscatelli@igag.cnr.it (M.M.); maurizio.simionato@igag.cnr.it (M.S.); gianpaolo.cavinato@igag.cnr.it (G.P.C.); francesco.stigliano@igag.cnr.it (F.S.)
2   Freelance Geologist, 00179 Rome, Italy; michele.dimasi2004@libero.it
*   Correspondence: cristina.disalvo@igag.cnr.it (C.D.S.); marco.mancini@igag.cnr.it (M.M.)

**Abstract:** This study presents the results of a research project financed by the Lazio Regional Government. The research focused on defining an integrated model of recent alluvial deposits in the Tiber River. To achieve this objective, geological boreholes were made to monitor the aquifer and in situ and laboratory tests were carried out. The data obtained were used to detail stratigraphic aspects and improve the comprehension of water circulation beneath the recent alluvial deposits of the Tiber River in the urban area of Rome, between the Ponte Milvio bridge and the Tiber Island. The stratigraphic intervals recognised in the boreholes were parameterised based on their litho-technical characteristics. The new data acquired, and integrated with existing data in the database of Institute of Environmental Geology and Geoengineering of the Italian National Research Council, made it possible to produce a three-dimensional model of the lithologies in the study area. The model of the subsoil, simplified for applied reasons, was described in hydrostratigraphic terms: three different lithotypes were subjected to piezometric levels monitoring. Finally, the research generated a numerical hydrological model in a steady state. In general, this study demonstrates how a numerical hydrogeological model calibrated by piezometric monitoring data can support the construction of a geological model, discarding or confirming certain hypotheses and suggesting other means of reconstructing sedimentary bodies.

**Keywords:** 3D geological modelling; groundwater models; incised valleys; Rome; alluvium

## 1. Introduction

This article describes the results of the activities defined in the research contract between the Institute of Environmental Geology and Geoengineering of the Italian National Research Council (IGAG-CNR) and Geoplanning-Servizi per il Territorio as part of the project "TIBER–Innovation in the field of geotechnics for the definition of tools, methods and procedures aimed at the building of a new model of subsoil (integrated model)," financed by the Lazio Regional Government. Additionally, it also presents the results obtained by a study successive to completion of the contract.

Objectives of the research project were the geological reconstruction and the creation of a three-dimensional lithotype model of the recent alluvial deposits of Tiber River in the urban area of Rome between the Ponte Milvio bridge and the Tiber Island. The lithotype model served as a base for hydrostratigraphic and numerical groundwater models.

In this framework, 3D models and lithological characterization are key aspects for groundwater modelling. In general, a geological model can be considered a three-dimensional (3D) spatial representation of the distribution of sediments and rocks below the ground surface. Specifically, a complex setting of subsurface terrains can imply important spatial

variations of terrain texture, cohesion, and geotechnical characters as well as hydraulic conductivity values. 3D voxel models can be tailor made to specific applications, i.e., mapping on-land exploitable aggregate resources [1] and clay resources [2]. A 3D representation of terrains geometry can also improve the capability to understand the geological hazards to which an urban area is generally exposed. Thus, if the effects of a 3D structure on the underground are important for solving a particular problem, a 3D model with associated hydrogeological, geotechnical, geomechanic, and seismic characteristics is required. A model with integrated rocks' geotechnical and hydrogeological properties (e.g., rock strength, compressibility, permeability, and porosity) is useful for engineering calculation in planning phases, helping to mitigate geological hazards which can affect new developments [3].

Furthermore, lithological characterization is important in groundwater modeling applications where the representation of spatial variability has a substantial influence on the behavior of the system [4]. Indeed, the spatial relationships between geological formations or different specific facies in a single formation are responsible for groundwater flowpaths' direction, and determine the conditions for aquifers' confining or communication conditions, groundwater storage, and release rates. Earlier large-scale 3D models are built to provide a basis for further groundwater flow models aimed at optimizing groundwater abstraction and management [5], or assessing if geological structures induce connectivity as barriers or conduits to groundwater flow [6].

On the geological reconstructions and modelling of the subsoil, various authors have compiled lists of the typical geological sequences in the centre of Rome, mainly following stratigraphic and depositional criteria [7–10], and also focusing on local geotechnical features [11], seismic properties [12–14], and hydrostratigraphic characteristics [15,16]. Many studies address the investigation of mechanisms inducing natural hazards such as settlements [17], flooding [18], and seismically induced amplification effects and permanent ground modifications [19,20].

Such works, however, did not include 3D lithological modelling specifically focused on groundwater level fluctuations. The innovative aspect of the present study is that the here-proposed models can be used to improve the understanding of the circulation of groundwater. For this purpose, geological drillholes and in situ boreholes were carried out, as well as laboratory tests and piezometric monitoring in two wells. It is worth to highlight the importance of coupling a detailed 3D model with groundwater monitoring, in order to correctly address the hydrogeological behaviour of each lithotype and their response to stresses such river floods. This was followed by a study of the stratigraphic levels recognised in the boreholes, which were parameterised based on their lithotechnical characters. The subsoil model, simplified for applied purposes, was then described in hydrostratigraphic terms, and a monitoring of the piezometric levels was applied in three different lithotypes. Finally, the results of the activities carried out under the project were successively utilised to produce a three-dimensional numerical hydrogeological model in stationary regime. The objective of the numerical model was to verify the conceptual model, comparing the values of hydraulic conductivity calibrated with theoretical values and analysing the spatial distribution of lithotypes in relation to the measured piezometric levels, thus providing the base for further transient-state model, at the scale of the site.

The study area constitutes an unusual zone in hydrogeological terms, as it represents an alluvial valley incised into the low permeability clays of the Vatican Hill. Owing to this characteristic, the alluvial aquifer-river system is not influenced by external factors, making it possible to study in detail the behaviour of alluvial sediments from external perturbations such as flood waves.

## 2. Geological and Hydrogeological Setting

### 2.1. Geology, Sedimentology and Fluvial Sequence Stratigraphy

The study area includes a portion of the alluvial valley of the Tiber River in the urban area of Rome. The Tiber has the second largest hydrographic catchment of any river in

Italy (17,000 km$^2$; Figure 1). The related hydrographic system began to develop during the late upper Pliocene-lower Pleistocene [21,22] and references therein). The lower course of Tiber is situated in the Roman Basin, an extensive tectonic-sedimentary basin from the Pliocene-Pleistocene epoch, where a relevant part of the sedimentary fill consists, since the late lower Pliocene (approx. 1.1 My), of a complex stacking of incised valleys filled with fluvial and coastal sediment and inter-fluvial deposits [23–25]. In the Roman Basin, the Tiber system recorded the fluvial responses to the concomitant actions of external controls, such as glacial-eustatic oscillations in sea level, volcanic activity from the neighbouring Volcanic Districts of the Sabatini Mountains and Alban Hills, sediment discharge from upstream sectors of the hydrographic basin, and regional uplift and the final phases of extensional tectonics involving the Tyrrhenian edge of the Apennines (Figure 1).

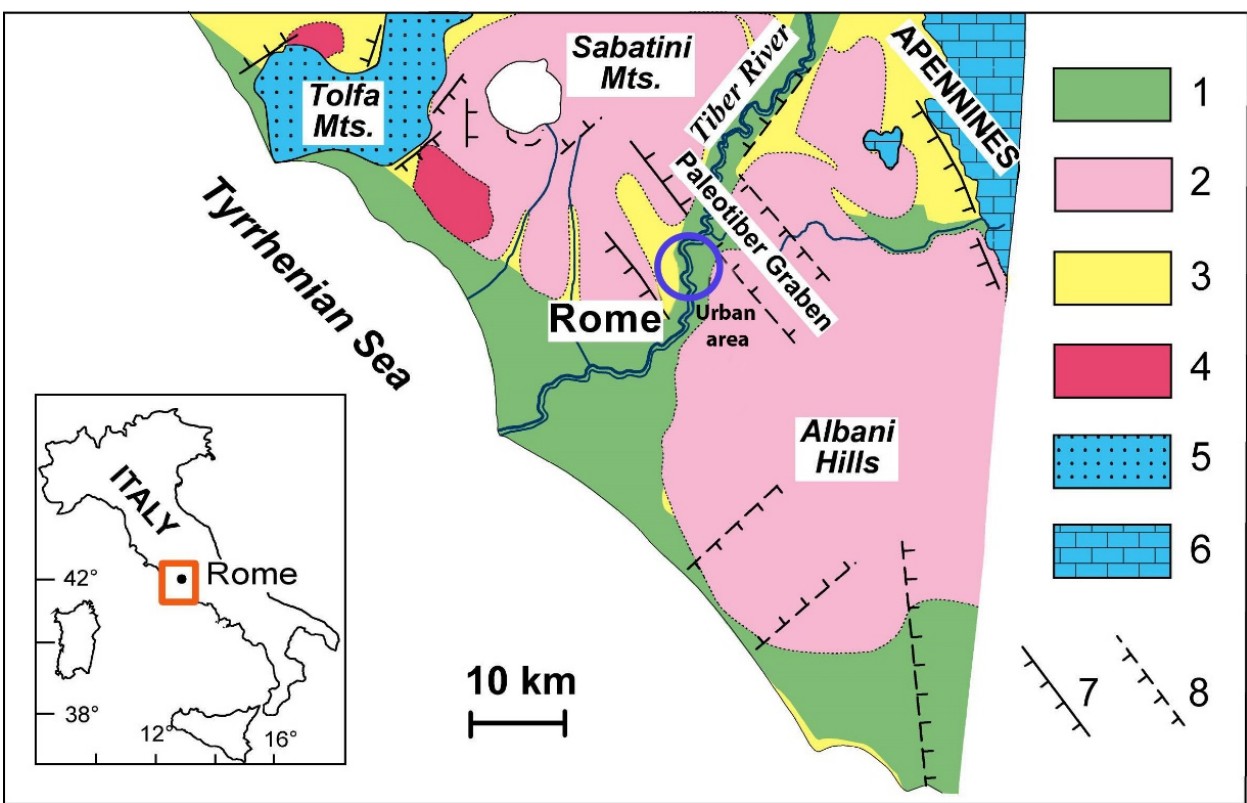

**Figure 1.** Geological setting of the study area and related depositional sequences. 1. Fluvial, coastal and shallow marine deposits; 2. Middle Pleistocene volcanites; 3. Coastal and shallow marine deposits; 4. Upper Pliocene volcanites; 5. Silicoclastic successions (Cretaceous-Eocene); 6. Carbonate successions (Trias-Miocene); 7. Normal fault; and 8. Buried normal fault.

On the glacial-eustatic oscillations in sea level, the most significant in the Roman Basin, in terms of preservation of associated sedimentary deposits and depositional architecture, is that relative to the last interglacial and glacial phase up to the recent sea level rise which occurred in the upper Pleistocene-Holocene (the last 116 ky). The sedimentary expression of this depositional cycle is represented by the Tiber River Depositional Sequence (TDS in [24], Figure 2a–c), a sequence with a high frequency and low range delimited by a clear unconformity surface (SB sequence boundary), characterised in general by a U-shape, transversal to the axis of the valley.

In the urban area of the Tiber (Figure 1), the TDS includes the incised Tiber valley. The latter is deeply set into the Pliocene-Pleistocene substratum and is filled with alluvial deposits, often up to a maximum of 60–70 m of thickness and deposited by the river in large part through mechanisms of backfilling [26] in response to the rising and high stand of the sea level. The ratio width/thickness of the filling (W/T ratio in [27]) is between 25 and 40 [28]. The fill deposits present a level-horizontal position. The stratigraphic architecture

of the incised valley fill was reconstructed through the examination and correlation of approximately 900 boreholes [24]. The relative stratigraphic descriptions are archived in the CNR-IGAG database, and the original data was provided largely by public entities and private companies. The stratigraphy of the boreholes was utilised to produce several correlation panels, or cross sections (Figure 2a–c). These panels were drawn to: (1) define the form of the valley and the limit of the Tiber River sequence at its base; (2) recognise the principal lithostratigraphic-sedimentologic components of the valley fill (such as channel bodies, floodplain, levee and crevasse splay deposits, organic and peaty layers, etc.); and (3) identify the most significant internal surfaces of the stratigraphic sequence (maximum flooding surface, and first transgressive surface) essential for correlating the fluvial deposits with the coastal, deltaic, and estuarine ones within the same TDS sequence. Additionally, three boreholes (S1, S2, and S3, with depths 59, 54, and 64 m, respectively) were drilled in the frame of the research project through the entire sequence as far as the base surfaces. These three boreholes were already described for defining their sedimentological and stratigraphic characters and were used as reference type-sections of the alluvial portion of TDS [24,25].

In the cited studies, a detailed facies analysis was made on these boreholes, keeping to the specific recommendations provided in the literature of fluvial deposits [29,30] which include the sequence boundary (SB), the first transgressive surface (ts), and the maximum flooding surface (mfs) (See Figure 2a–c; after [28]). These key surfaces identify substantial changes to the general configuration of the facies in the various depositional environments in relation to the variations between the space of accommodation and the sedimentation rate, driven by: the variations in sea level, the hydraulics of the system, the compaction of alluvial terrains, tectonics, climate, and the sedimentary supply [26,27,31–35]. These surfaces of stratigraphic correlation made it possible to vertically subdivide the alluvial terrains into three portions with a stratigraphic continuity, i.e., the systems tracts, laterally continuous to those recognised in the deltaic sector of the Tiber Sequence [24,36]. The oldest deposits in the sequence correspond with the early lowstand or falling stage, systems tract (ELST), featured by rare remnants of buried terraced sediments, deposited in a fluvial environment during the lengthy phase of a lowering of the base level (between approx. 116 and 26 ky (see cross Section 4, Figure 2c).

These terraces, up to 20 m thick, consist of a fining-upward succession of gravels and sands that become clays and overconsolidated silts. The basal lower portion of the sequence, corresponding with the late lowstand systems tract (LST, between 26 and 14 ky), is recorded between the sequence boundary SB and the first transgressive surface ts, and between 8–10 m thick. This portion of the sequence is associated with amalgamate gravely deposits, at the bottom of the incised valley (see also [24]). The gravels consist of well-rounded stones with a maximum diameter of 8 cm, formed of limestone, flint, and more rarely of sandstone and pyroclastics. The matrix consists of coarse sands rich in ferromagnesian minerals.

The intermediate portion of the sequence, corresponding with the transgressive systems tract (TST), is up to 40 m thick and located between the transgressive surface, ts, and the maximum flooding surface, mfs, both with a planar geometry. The base of the TST consists of gravels fining-upward into medium-coarse sands of braided river environment, with a maximum thickness of 10 m, and to the sides of overconsolidated and pedogenically modified, floodplain clayey deposits. The overlaying sands define the filing of channel bodies, each of them up to 6–8 m thick and 200 m wide, with a vertical fining-upward of lithofacies from granules to fine silty sands. The channel bodies are attributed to a meandering fluvial environment and are laterally bounded by largely unconsolidated grey clays, organic-and peaty-rich, and typical of a poorly drained, swampy floodplain. The upper portion of the sequence is 20 m thick and positioned between the mfs and the current topographic surface, corresponding with the highstand systems tract (HST). In this portion of the sequence, the sandy channelized bodies tend to expand laterally to 600 m, as a consequence of the reduction in the space of accommodation and channel clustering, while the floodplain deposits consist of overconsolidated and highly pedogenised silts and clays.

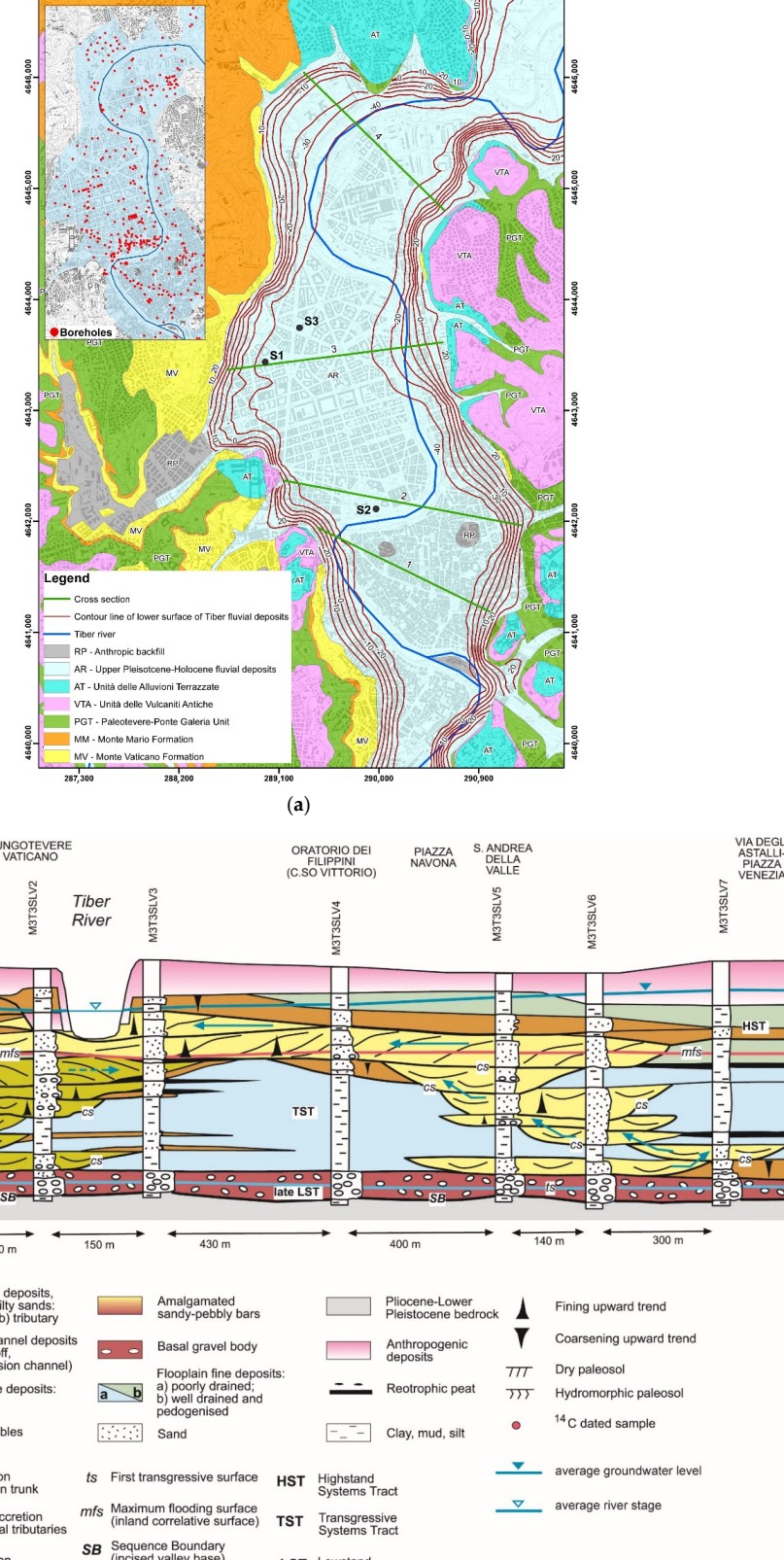

**Figure 2.** *Cont.*

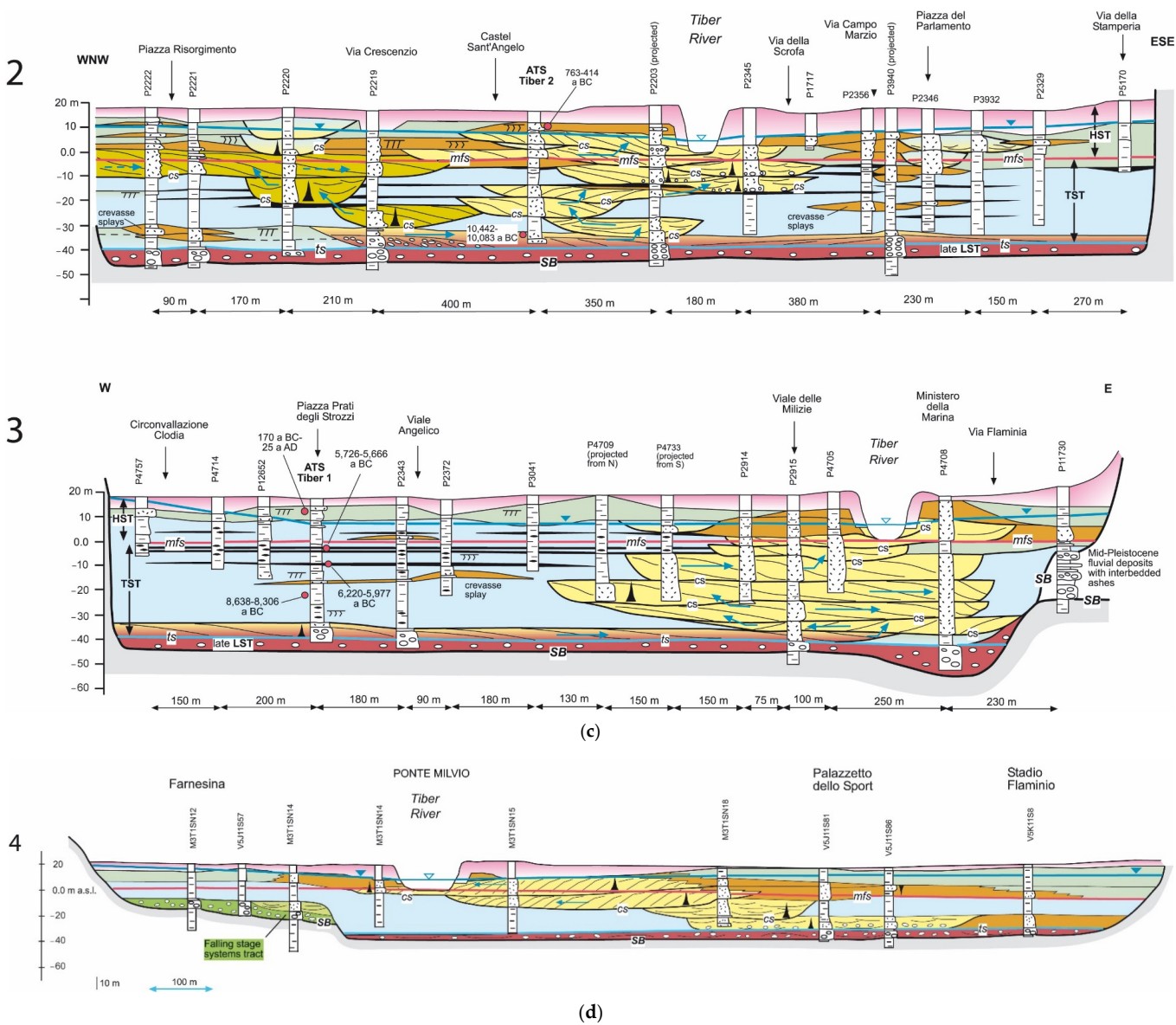

**Figure 2.** (**a**) Geological map of the study area, location of S1-S2-S3 boreholes, and trace of cross sections of (**b**)–(**d**). The basal surface of Tiber River depositional sequence is also mapped (contour lines every 10 m). Reference system: WGS 84/UTM zone 33N. (**b**) Cross Section 1 (Piazza Pia- Piazza Venezia) through the Tiber River depositional sequence (after [25]). Groundwater level data are from monitored boreholes and literature data ([37]). (**c**) Cross Sections 2 and 3 through the Tiber River depositional sequence: 2: Piazza Risorgimento-Fontana di Trevi. 3: Prati-Flaminio. (**d**) Cross Section 4: Farnesina-Stadio Flaminio. Traces in (**a**); legend in (**b**).

The definition of the lithofacies within the sequence and their spatial variability derive principally from the analyses and correlation of well data. This was accompanied by the mapping of the lithofacies at various depths, which permits the identification in plan of the principal associated paleo-geographic elements and sub-environments (such as active braided channel areas, or channel belt areas or active channel-levee, drained and undrained floodplain) and to reconstruct their areal distribution.

### 2.2. General Hydrogeological Setting

In the area of Rome underground flow is directed by high piezometries, located on the slopes of the Sabatini Hills and Alban Hills, toward the Tiber valley, which crosses the city in the NE-SW direction (Figure 3); the underground waters run initially within the

volcanic and pre-volcanic sedimentary units, before flowing into the alluvial terrains of the Tiber valley and being drained toward the Tyrrhenian Sea.

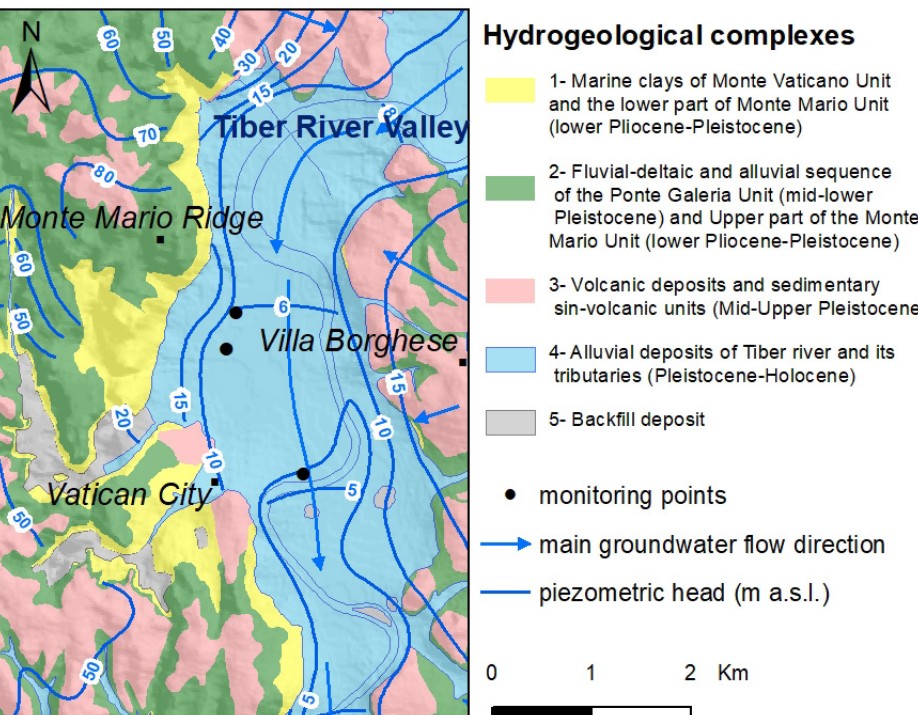

**Figure 3.** Hydrogeological complexes in the study area. The boreholes S2 and S3 are equipped with piezometers for groundwater monitoring.

The most superficial aquifer is hosted in the volcanic formations and the uppermost part of the sedimentary succession of the "Ponte Galeria" Unit ([7,9,38]), corresponding with the Santa Cecilia Unit ([39]). The remaining portion of the Ponte Galeria Unit in turn presents more overlapping aquifers, consisting of sandy-gravely layers bounded by clayey interlayers. Overall, circulation is supported at the base by the impermeable clays of the Monte Vaticano formation ([7,9,15,40]).

The Tiber valley thus constitutes the receiving hydrogeological unit for subterranean waters. It is characterised by different depositional facies that determine a notable heterogeneity in values of permeability (Table 1).

The elevation of the water table at the edges of the valley varies between 15 and 10 m (decreasing from north to south) and presents a low hydraulic gradient (i = 0.002, measured in the historical centre [41,42]). The recharging of the Tiber valley aquifer is a result of stratigraphic contact with the units into which the valley is incised. The greatest hydraulic transfer is hypothesised where the fluvial valley is incised into the sedimentary deposits of the Ponte Galeria Unit. On the contrary, where the valley is incised into the low permeability sediments of the Monte Vaticano and Monte Mario Clays, the hydraulic transfer can be considered negligible. The elevation of the water table is strongly influenced by the level of the Tiber River; under normal flow conditions, the Tiber acts primarily as a receiver from the aquifer (i.e., acts as a gaining stream); under flood conditions, the Tiber feeds the aquifer (i.e., acts as a losing stream).

**Table 1.** Hydrogeological parameterisation of the complexes (from [16]).

| Stratigraphic Frame | | | Complex Code (Di Salvo et al., 2012) | Hydrogeological Compex | | Range of Kx (m/d) | Variance $\sigma^2_{\log K}$ | n of Tests |
|---|---|---|---|---|---|---|---|---|
| Anthropic backfill | | | RP | Complex 5 | | 0.04–20 | 0.91 | 5 |
| (Holocene) | | | | | | | | |
| Tiber alluvium—Clay and silty clay | | | AR | Complex 4 | 4a | 0.000034–0.5 | 0.84 | 17 |
| Tiber alluvium—Sand | | | | | 4b | 0.032–43.2 | 0.65 | 41 |
| Tiber alluvium—Clay with peat | | | | | 4c | 0.000017–0.017 | 0.1 | 5 |
| Tiber alluvium—Silty, Sandy gravel | | | | | 4d | 0.003–6.5 | 0.83 | 23 |
| (Holocene) | | | | | | | | |
| Volcanic units | Ancient alluvium formation | Terraced alluvium formation | VTA | Complex 3 | | 0.172–6.048 | 0.9 | 9 |
| (Middle-Upper Pleistocene) | | | | | | | | |
| "Fosso della Crescenza" Unit | | | PGT | Complex 2 | | 0.000397–0.292 | 0.83 | 50 |
| (Lower-Middle Pleistocene) | | | | | | | | |
| "Monte Mario" Unit | | | MM—upper portion | | | 0.1 | 0.13 | 3 |
| (Lower Pleistocene) | | | MM—lower portion | Complex 1 | | 0.0001–0.01 | 0.7 | 3 |
| "Argille di Monte Vaticano" Unit | | | MV | | | | | |
| (Upper Pliocene) | | | | | | | | |

Description of Hydrogeological Complexes:

This section describes the principal hydrogeological complexes in Rome according to the classification provided in [16] (Figure 3). Table 1 lists the values of hydraulic conductivity.

Complex 1: includes the marine clays of the Monte Vaticano Unit and the lower part of the Monte Mario Unit (the so-called "Membro di Farneto" in [22]) from the lower Pliocene-Pleistocene. Its permeability is very low (0.001–0.0001 m/d); the complex has an aquiclude function for the study area.

Complex 2: corresponds with the fluvial-deltaic and alluvial sequence of the Ponte Galeria Unit of the mid-lower Pleistocene. It also includes the upper part of the Monte Mario Unit. This complex is characterised by a notable lithologic heterogeneity; the range of permeability is between 0.01 and 0.14 m/d. Where it is covered by a variable thickness of volcanic deposits, it may host neighbouring aquifers.

Complex 3: includes all the volcanic deposits and sedimentary sin-volcanic units of the Mid-Upper Pleistocene, and presents highly variable values of permeability (from 0.1 up to 6 m/d). The highest values of permeability are characteristic of non-lithified pyroclastic deposits and highly fissured tuffaceous rock formations: both of these lithologies host the most superficial, phreatic aquifer. This complex covers practically the entire urban area, with the exception of the Monte Mario chain and the alluvial valleys of the Tiber and Aniene rivers.

Complex 4: includes the deposits that fill the Tiber valley and its tributaries, datable to the upper Pleistocene-Holocene. They consist of silty clays and silty-sandy deposits, atop a base level of sandy and silty gravels. Internally, it features four distinct sub-complexes, corresponding with different depositional environments: coarse sediments (high permeability) are related to a high energy sedimentary regime, while silty-clayey sediments are connected with low energy environments such as floodplains or fluvial-palustrine environments.

*Complex 4a:* is the most superficial unit, characterised by clays and silty clays.

*Complex 4b*: consists of fine and coarse sands, often with a silty matrix. The sands, deposited in a fluvial channel environment, are surrounded by the clays of complex 4a.

*Complex 4c*: consists of grey clays with organic matter.

*Complex 4d*: this sub-complex consists of a discontinuous bed of silty-sandy gravels varying between 0 and 30 m in thickness. This deposit is related to high energy flu-

vial environments and characterised by highly variable values of hydraulic conductivity (0.03–6.5 m/d), which depend on the matrix of the deposit.

Complex 5: The urban territory is covered by a practically continuous cover of backfill. Given its average values of permeability (0.1–0.01 m/d), this deposit can function as an aquifer when it reaches significant thicknesses (as in the historical centre, where it surpasses 15 m).

The detection of different lithotypes in the alluvial valley, which are associated to ranges of hydraulic conductivity, allows the description of the alluvial complex in hydrostratigraphic terms.

Indeed, within alluvial deposits, the hydraulic relations among diverse complexes, as well as aquifer-river exchanges, are highly variable and a function of local stratigraphy. Measuring campaigns demonstrate that the aquifers hosted in the complexes of the base sands and gravels (complexes 4b and 4d) present a very similar static level, despite being located at clearly different depths. In some cases, the level of the base gravels (at a depth of approximately 50 m below the ground [37,43]) surpasses the level of the more superficial sandy aquifer by roughly 50 cm: therefore, in this area the two complexes show a diverse piezometric head level. In the areas in which complexes 4b and 4d are in stratigraphic contact, the piezometric levels are assimilable. For a detailed investigation of this type of hydraulic relation, this study included the planning of a continuous monitoring of the alluvial complex.

## 3. Materials and Methods

The workflow of Figure 4 summarizes the steps of the study, which are described in the sections below.

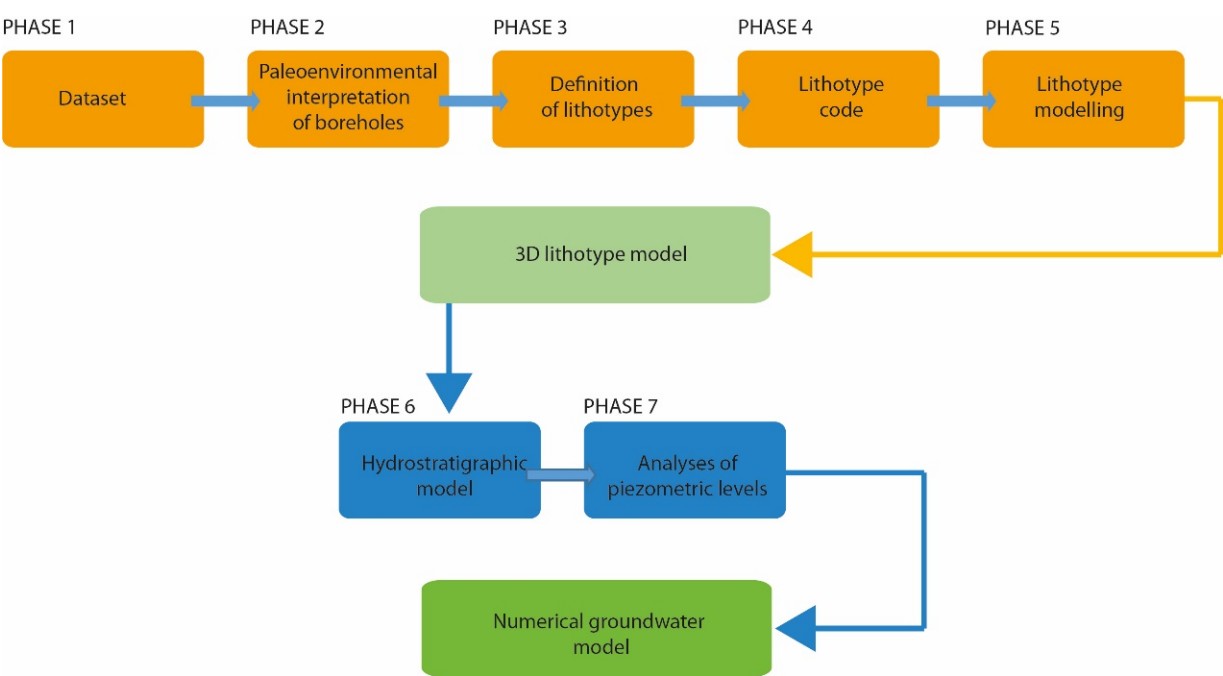

**Figure 4.** Workflow showing the phases of the 3D lithotype model development.

### 3.1. Boreholes Analyses and Correlation

Differently from the facies interpretation presented in detail in [24,28], here a new interpretation is presented in geo-lithological units. Stratigraphic surfaces of particular significance in terms of correlation were identified within the valley fill. Table 2 lists the main features of the drilled boreholes.

**Table 2.** Coordinates (UTM WGS84, 33N), elevation, and depth of drilled boreholes. The distance between the wells and the riverbank was calculated orthogonally to the axis of the river. The distance between the well and the mouth of the river, along both the median of the alluvial valley and the river channel, was also calculated.

| Borehole | X (UTM WGS84) | Y (UTM WGS84) | Elevation (m a.s.l.) | Depth | Borehole Distance from the Riverbank (km) | River Mouth Distance along the Alluvial Valley (km) | River Mouth Distance along the River (km) |
|---|---|---|---|---|---|---|---|
| S1 | 288,892.34 | 4,643,538.26 | 17.0 | 59 | 1.225 | 29.29 | 41.45 |
| S2 | 289,991.13 | 4,642,103.57 | 18.2 | 53.5 | 0.066 | 29.00 | 39.37 |
| S3 | 289,265.5 | 4,643,739.93 | 17.6 | 65 | 0.987 | 31.00 | 41.2 |

Sedimentological, mineralogical, and chemical analyses were made by extracting samples every 20 cm of perforation. The samples were tested to date and to define a set of physical parameters such as granulometry and mineral composition, and to establish dates for the age of the sample. In particular, the following tests were made:

- sedimentological (granulometric laser analyses)
- mineralogical (diffractometric analyses)
- chemical (definition of the content of water and crystallisation up to 200 °C, content of oxidisable organic matter up to 600 °C, inorganic carbonates up to 850 °C)
- micropaleontological (calculation of the fossiliferous content of the lithotypes)
- radiometric (C14 dating of organic matter).

Based on the results obtained, and the fossils and sedimentary structures present, it was possible to reconstruct the environment of sedimentation ([24,25,28]).

*3.2. Geolithological Mapping*

The availability of geognostic tests, their homogenous distribution across the entire study area, and the good quality of the descriptions of the stratigraphic intervals traversed made it possible to create a geolithological model of the recent deposits of the Tiber River. The geolithological model served as a base for hydrostratigraphic and numerical groundwater models in the following steps. The correlation between boreholes was used to define the associations of lithofacies. The mapping of the lithofacies, at diverse depths, permitted the reconstruction of their areal dimension. The associations of lithofacies were then used to extract the lithotypes that are lithological units used in the process of geotechnical and hydrogeological characterisation. The lithotypes were identified by combining lithofacies with similar granulometric-textural characters. Given the direct correlation that exists between the association of lithofacies and lithotypes, the geometric structure of the lithotypes coincides in fact with that of the lithofacies.

Hence, from the spatial distribution of the lithofacies it is also possible to visualise the distribution of the lithotypes with their associated physical-mechanical parameters. In other words, this is a manner for representing and visualising the lithological domains with homogenous physical and mechanical characteristics. The reconstruction of the geometries of these lithological bodies becomes a fundamental base geological element for diverse applied questions which may present themselves in urban areas (settlement, variations in permeability, etc.).

Beginning with this process, it was possible to create a three-dimensional model of the lithotypes recognised in the Tiber valley, in the sector between the Ponte Milvio bridge to the north and the Tiber Island to the south.

The creation of the 3D model was carried out with the methodology described in the following sections.

3.2.1. Codification of Each Stratigraphic Interval Traversed by Boreholes and Recognised Lithotypes

The first step toward the realisation of the three-dimensional model of the recent alluvial deposits was the codification of the boreholes, by means of the lithotypes defined

in Table 3 and the stratigraphic framework of Figure 5. The three-dimensional modelling software utilised (RockWorks 15, RockWare®) requires the definition of numerical values to interpolate the diverse lithologies in the model (Table 3). Thus, it was necessary to define numerical codes for the lithologies present, used to codify the diverse stratigraphic horizons encountered by the boreholes.

**Table 3.** Lithotypes and associated codes.

| Lithotype | Code |
| --- | --- |
| Gravels and gravelly sands | 1 |
| Sands and sandy silts | 2 |
| Clay and clayey inorganic silt | 3 |
| Clay and organic clayey silt | 4 |

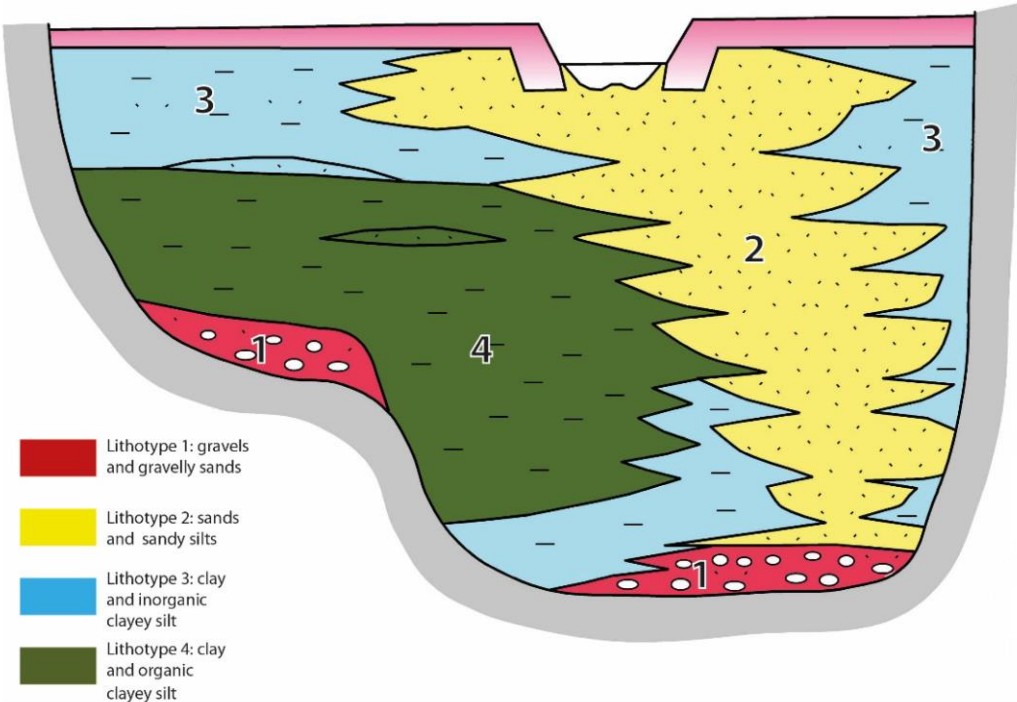

**Figure 5.** Descriptive section of the distribution of lithotypes in the alluvial body.

### 3.2.2. Flattening of Bodies

Geometries of the depositional surface of the sedimentary bodies depend on the conditions existing at the moment of deposition, while the current position of the bodies depends on eventual tectonic movements successive to the deposition. Hence, it is not a given that the surface of deposition and the current position coincide. In this study, the surfaces of deposition of the alluvial terrains were correlated parallel to each other and inclined toward the coastline, with the same angle of inclination of the current topographic surface.

If we were to operate a horizontal cut at a given depth in the Tiber River depositional sequence, we would intercept diverse stratigraphic intervals at the same depth, younger toward the valley and older uphill. Therefore, all of the depositional surfaces, as well as the current topographic surface, were rotated so that the horizontal cut lies inside the same isochrone.

This required the definition of the so-called reference level, which is the level of flattening, and, if necessary, to re-establish the horizontal continuity of the lithotypes.

Having identified the reference level, the domain of the study was deformed (this operation is known as flattening) through geometric transformations of the coordinates, with return the reference level to a horizontal plane. In this case study, the reference level

is essentially linked to the topographic structure of the area: a depositional surface sloping toward the coastline and with a slope equal to the average gradient of the current axis of the river. The geometry of the deposits was restored by transforming the coordinates through an operation of rotation.

### 3.2.3. Mapping of the Diverse Depths of Lithotypes and Definition of the Spatial Resolution

The next step was a mapping at diverse depths, every 5 m, of the recognised lithologies in order to identify in plan the principal associated geomorphological elements and environments (active channel-levee areas, and drained and undrained floodplain; Figure 6) and reconstruct its areal distribution. This was followed by a process of rasterising the surfaces at a cell resolution of 50 m (Figure 6).

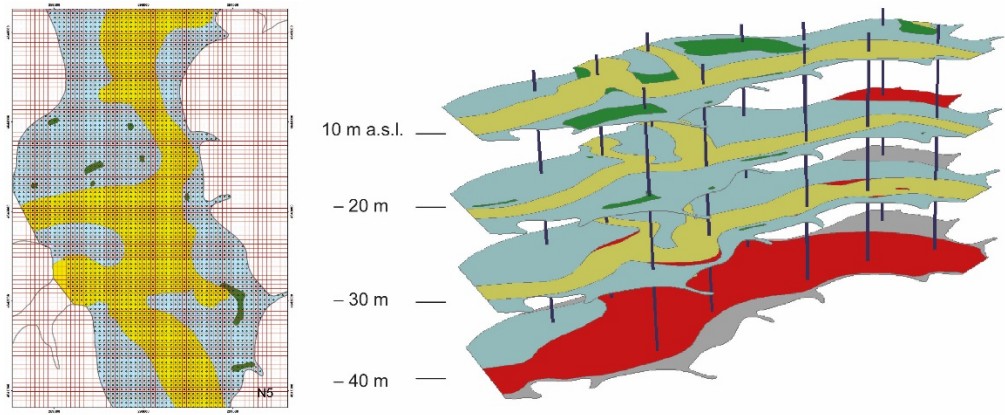

**Figure 6.** Rasterization of the model with a cell resolution of 50 m, on the left. On the right, reconstruction of paleoenvironments in the Tiber River valley. Active channel and channel-levee areas (red for gravels, and yellow for sands and silty sands) and drained and undrained floodplain (blue for clays, and green for peats) are mapped at different depths. In gray are the lithotypes of the substrate. Vertical boreholes (black lines) used to constrain the model are also reported.

This cell value was determined through a comparison with the average dimensions of the sedimentary bodies to be modelled. In the model to be created, the bodies present an average width of 100–200 m, and therefore the selected resolution was considered suitable for the representation of the different bodies. A lower resolution (and a consequent large cell dimension) would lead to a loss of information.

Each lithotype was associated with a numerical code, in accordance with Table 3.

### 3.3. D Geolithological Model

The mapping of the lithotypes every 5 m facilitates an almost three-dimensional reconstruction of the sedimentary bodies, which can be utilised as an instrument for further geotechnical and hydrogeological analyses. As described in Section 3.2, the spatial distribution of the lithotypes with the associated geotechnical (physical-mechanical) and hydrogeological parameters mimics the distribution of lithofacies filling the river valley. This permits the identification of relatively homogenous lithological domains in the subsoil, in terms of associated geotechnical and hydrogeological properties. Regarding the relations between the sedimentary features and hydrogeology, it is well known that the lateral-verticality of the lithofacies and depositional elements in the subsoil can condition hydraulic connectivity in alluvial terrains ([44,45]). Therefore, a proper reconstruction of the stratigraphic architecture in alluvial contexts, in particular for incised valleys, is fundamental for the geometric characterisation of aquifers.

The three-dimensional model was therefore created using the software Rockworks produced by Rockware ([46]). Rockworks software algorithms can be used to interpolate continuous stratigraphic surfaces that, when stacked, form a 3D stratigraphic model. Other functions enable the user to create individual logs or multi-log cross sections, fence-

diagrams, and 2D and 3D maps. Furthermore, it provides some GIS processing tools as well as import/export capabilities in various formats ([47]). The resulting model is a numeric file containing a list of points (X, Y, and Z) regularly distanced and interpolated based on a value that represents the parameter to be modelled; in the case examined here, the lithology.

The distance between points determines the measure of the "voxel" that, in fact, represents the resolution returned by the model. The size of the voxel or resolution of the model, in general, is calculated automatically by the software based on the distribution of the data present in the study area. In the case examined here, as described in Section 3.1, the voxels were based on the values of 50 m (X) × 50 m (Y) × 5 m (Z).

The resulting model was successively cut vertically and horizontally to obtain profiles, sections, fence diagrams, and slices at different depths. The model of the lithologies was reconstructed utilising the following methodology:

1. rasterization using a 50 m × 50 m cell of each paleogeographic map;
2. assignment at the centroid of each cell of the corresponding lithology;
3. reconstruction, for each vertical passing through the diverse centroids, of the litho-types encountered at diverse depths.

This makes it possible for the entire study area to define the lithologies present along a series of horizontal layers, from 15 m a.s.l. to −40 m a.s.l., with 5 m intervals. The base data are input into the software using Rockworks Project database which requires the mandatory compiling of the following fields: Borehole Name; Easting, Northing; Elevation, and TD -Total Depth.

Successively, for each single borehole, it was necessary to compile the Lithology Datasheets Table. This table requires that the following are defined for each lithological horizon in the borehole:

- depth from grade to the top of the layer (Depth to top)
- depth from grade to the bottom of the layer (Depth to Base)
- lithological type (Keyword) connected with the Lithology Types Tables, listing the numerical codes defined for the diverse lithologies to be modelled.

After assigning the codes and patterns to the lithologies, using the "Lithology Model", it was possible to model the geometries of the diverse lithological bodies identified in three dimensions. As an algorithm of interpolation of data, the choice was made to use the Nearest Neighbours method. This method of interpolation makes it possible to assign to the node of each Voxel the value of the closest Voxel. This method has the advantage of honouring the data input into the model, though it has the disadvantage of returning a model of blocks, and thus the lithologies do not present the nuances found in nature, but represent them with brusque interruptions.

Restrictions or constraints were then applied to the model to fix the modelling of the lithologies in space (Figure 7). The following constraints were utilised: for the upper surface the base of the backfill was applied, while for the lower surface the cover of the Monte Vaticano Formation was applied (Table 1).

### 3.4. Hydrogeological Monitoring

For the monitoring of aquifer levels as described, two open standpipe piezometers were installed in boreholes S3 (S3-OS) and S2 (S2-OS), and an electric Vibrating Wire in piezometer S3 (S3-CV). The piezometers facilitate the monitoring of complexes 4a (clays), 4b (sands), and 4d (gravels in a sandy-silty matrix), shown in Table 4.

Dataloggers for continuous head level monitoring were installed in the open standpipe piezometers. Additionally, manual measurements were taken periodically using a water level meter (phreatimeter). The dataloggers of both the open standpipe piezometers and the electric piezometer were programmed to take hourly measurements from September 2014 to May 2015.

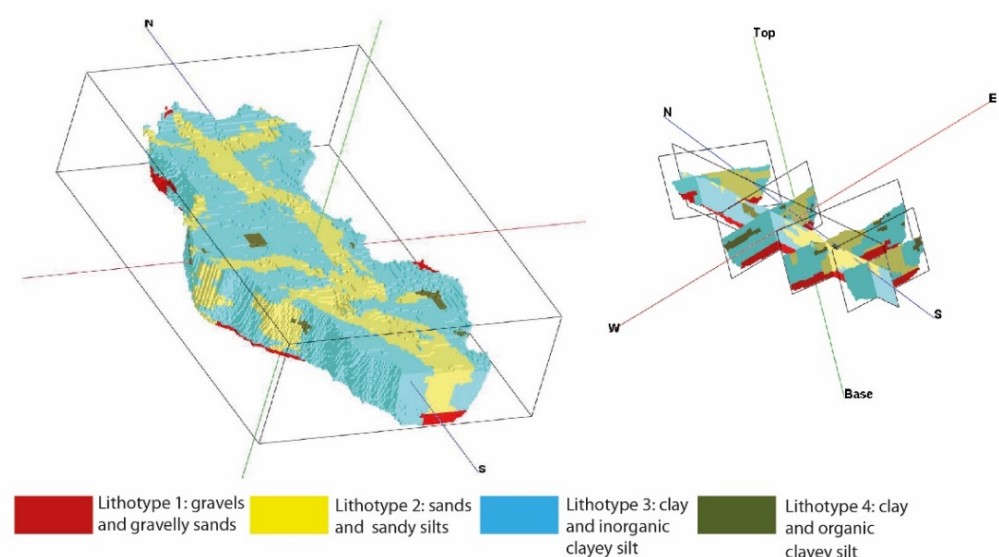

**Figure 7.** 3D model (**left side**) of the Tiber River depositional sequence and model cut into cross sections (**right side**).

**Table 4.** List of piezometer types, monitored complex, and depth of the screened interval or monitoring cell.

| Borehole | Piezometer Type | Monitored Complex | Depth of the Fissured Interval or Monitoring Cell (m from Ground) |
|---|---|---|---|
| S3 | Open standpipe-OS | 4d | 30–44 |
| S2 | Open standpipe-OS | 4b | 50–58 |
| S3 | Vibrating Wire-VW | 4a | 27 |

*3.5. The Numerical Groundwater Model*

The 3D reconstruction of the lithotypes was used to set up a steady state numerical groundwater model with the MODFLOW 2000 code ([48]) and the Groundwater Vistas 6 graphical interface.

The objective of the model was to verify the conceptual model, comparing the values of hydraulic conductivity calibrated with theoretical values.

The model was implemented in the portion with the greatest density of piezometric control points (targets); other than the monitoring piezometers of the present study, they also include the piezometric data collected between 1991–2010, selected and extracted from the UrbiSIT-LINQ database ([49]).

The numerical model area covers a portion of the geological model, for a total area of 6,336,000 m$^2$ (66 rows, and 60 columns of dimensions 40 × 40 m$^2$), and was set up with seven layers. The initial recharge was assigned uniformly to the domain of the model, considering the average values of effective infiltration in the urban area of Rome (0.00292 m/d [49]). The recharge value was then subjected to calibration.

River boundary conditions (Cauchy type) are used to simulate Tiber River in layer 4; the required input of riverbed hydraulic conductivity was set uniformly equal to 1 m/day, which is a reasonable average value for sandy–clayey riverbed sediments ([50,51] and references therein).

Constant head boundary conditions were set along the eastern boundary in layer 1 in order to reproduce the observed head field at the model boundary.

The initial values of horizontal hydraulic conductivity (Kx = Ky) were assigned to the lithotypes in accordance with Table 1; the values of vertical conductivity were assigned as 1/10 the value of Kx and Ky.

The model was calibrated using 86 observation points (targets) using the PEST calibration software.

## 4. Results

### 4.1. Results of Boreholes Interpretation

In terms of lithofacies, boreholes S1, S2, and S3 are representative of limey-clayey and sandy channel deposits (Figure 8). In particular, boreholes S1 and S3 present limey-clayey terrains belonging to the floodplain facies (with channel sands and gravels at the base), while borehole S2 is representative of sandy channel deposits.

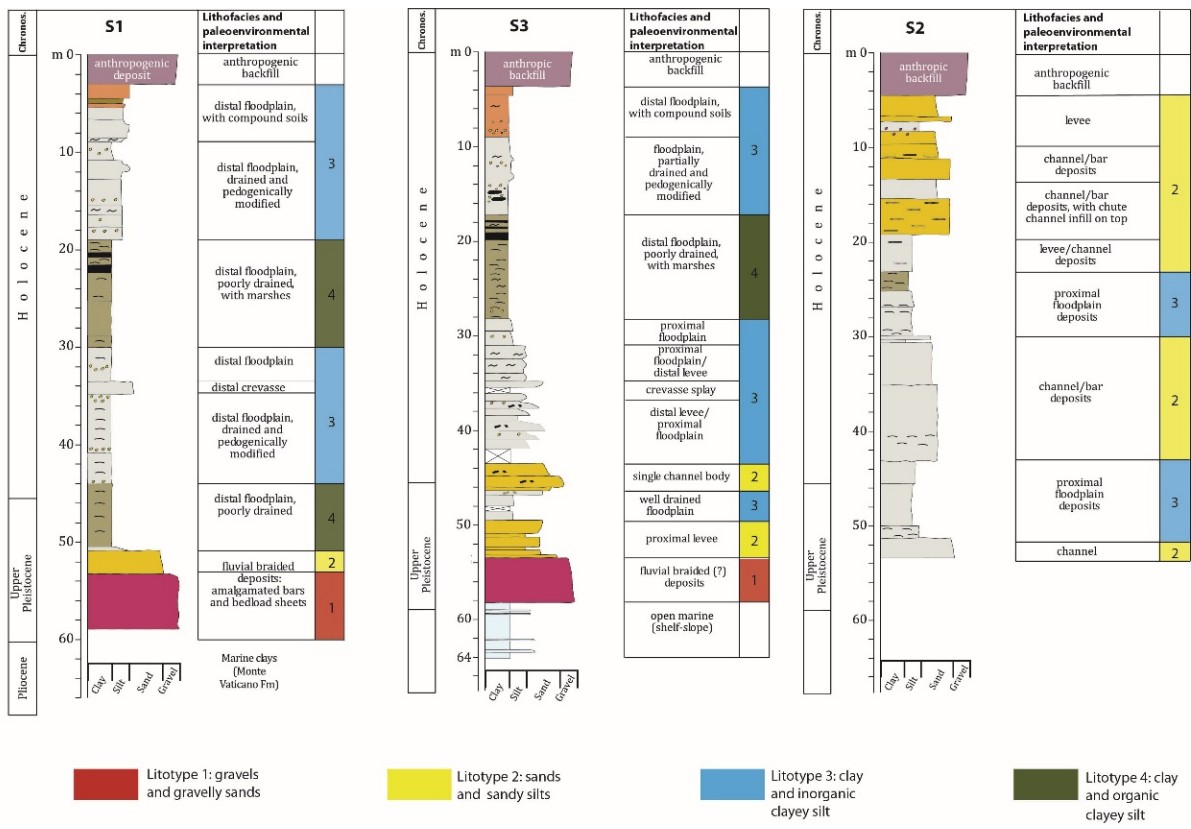

**Figure 8.** Stratigraphic-sedimentologic logs of boreholes S1, S2, and S3 (after [8]).

Table 5 illustrates the correlation between the associations of lithofacies, the depositional environments, and the reference lithotypes for the integrated model of the subsoil. In particular, the lithotypes are lithologic units for the geotechnical and hydrogeological characterisation of the subsoil and are identified by combining the associations of lithofacies with similar granulometric-textural features. There is also a good correspondence among the lithotypes, always clearly identifiable from an attentive reading of the descriptions of the borehole data, and the depositional environments, defined by the associations of lithofacies based principally on sedimentary and pedogenic processes. The stratigraphic correlation between lithologies and lithotypes carried out using the borehole tests within the framework defined in Table 5 made it possible to provide very restrictive stratigraphic and sedimentologic limits for the correlation between tests, for the reconstruction of the phases that filled the valley and the calculation of the aggradation rates.

**Table 5.** Comparison between associations of lithofacies, depositional environments, and reference lithotypes for the integrated model.

| Associations of Lithofacies | Depositional Environments | Lithotypes |
|---|---|---|
| Active channel, gravels and fluvial sands (Gs) | *Gravel bed braided river* | Gravels, with gravelly sands |
| Active channel, medium-fine sized sands and silts (Smf); crevasse splay, fine-sized sands and silts (Scr); abandoned channel, fine-sized sands, silts and clays (Sch); levee, heterolithic alternations of sands and muds (Sp) | *Channel belt* | Sands and sandy silts (the lithotype in question also includes deposits not strictly linked to the channel-bank system, but also the sandy-silty deposits of *overbanks* linked to the crevasse splays but, however, in general adjacent to the original *channel belt*) |
| Drained floodplain muds (Dp). Undrained floodplain muds (Dp). Facies belonging to formations previous to the most recent climatic cycle | *Well drained floodplain* *Undrained floodplain; marsh; peat fen* *Geological substrate* | Clays and inorganic clayey silts Clays and organic clayey silts Marly clays, sands, gravels, and pyroclastics. |

*4.2. Results of the Geolithological and 3D Models*

Results of the geolithologic model and 3D voxel model show the distribution of facies and, thus, the hydraulic conductivities in the modelled area, which vary spatially. The bulk of the alluvial valley is constituted by complex 4a (clay and inorganic clayey silt), typical of a well-drained floodplain. The sandy and sandy-silts lithotype (complex 4b) runs in the middle of the valley, indicating the position of the main fluvial channel at different depths, and its tributaries, both in the right and left bank. Sporadically, lenses of clay and organic clayey silt (complex 4c) appear laterally to complex 4b, in the first 20 m from the surface, indicating areas of undrained and/or mash floodplain surrounding the channel belt. Finally, the model reproduces a continuous layer of silty-sandy gravel (complex 4d) with a thickness ranging from 8 to 10 m, standing at the base of the sequence. Thus, heterogeneity of the aquifer system can be represented by a spatial variation in hydraulic conductivity ranging between that of clay and gravel. Indeed, the lithological variation implies that hydraulic conductivities vary spatially and control groundwater flow regime. The central channel belt of complex 4b, due to its elongated geometry and high hydraulic conductivity, can facilitate preferential flowpaths, which could also control contaminant transport; being in direct contact with the Tiber River, it ensures the exchange between groundwater and surface water. The gravel layer at the bottom of the sequence represents a continuous, high-permeability layer, capable of exchanging groundwater with the complex 4b, with which is connected. Differently from the sandy channel belt, it is widely constrained at its top by the complex 4a and by the complex 1 at its bottom; thus, being largely confined by low-permeability terrains, it can host a pressured aquifer. These results help to complete the interpretation of groundwater monitoring as described in the discussions.

*4.3. Results of Chemical-Physical and Level Monitoring*

The results demonstrate that the oscillations in the piezometric level of the gravely and sandy lithotypes are strictly dependent on the flow of the river (Figures 9 and 10). The level of the Tiber (recorded at the Ripetta station) is subject to oscillations of various periods, including oscillations in tides and flood waves. The oscillations with the shortest period are the result of tide oscillations; a complete tide oscillation in the Mediterranean Sea lasts roughly 12 h and 25 min. As a consequence, over the course of 24 h there are two high points and two low points, with oscillations in the order of 0.25 m and a maximum range of 0.43 m. The head level in piezometers S2-OS and S3-OS is influenced by both the peaks of fluvial flooding and the oscillations caused by tide cycles. The amplitude of the oscillations is minor with respect to the level of the Tiber. Furthermore, the dampening of the amplitude of the oscillations is greater for S3-OS than for S2-OS. The level in piezometer S2-OS is

always greater than the level of the Tiber, except during flooding, when an inversion occurs and the river passes from gaining to losing.

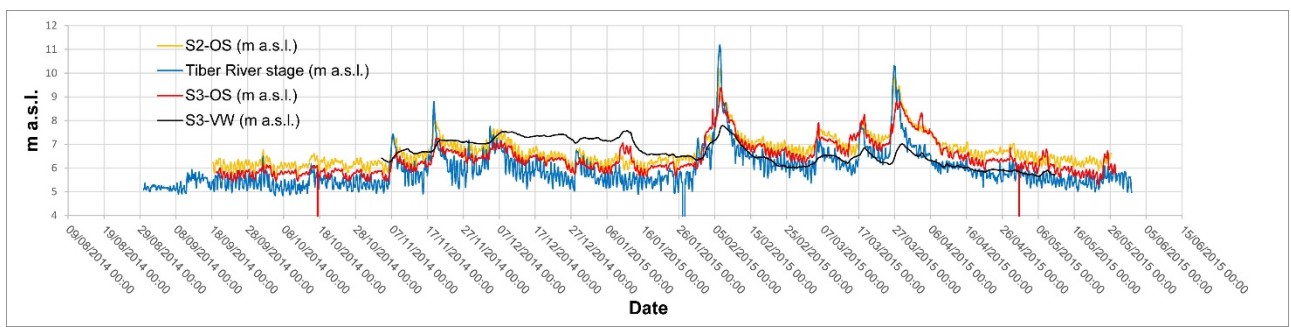

**Figure 9.** Recorded head level in the three piezometers and comparison with the Tiber River.

**Figure 10.** Recorded head level and comparison with rainfall and Tiber stage in Open Standpipe piezometers S2 (**A**), Open Standpipe S3 (**B**), and Vibrating wire S3 (**C**).

The level in piezometer S3-OS is on average lower than the level of the Tiber, though in some cases it is greater. The short-period changes in levels (seen as "peaks" in Figures 9 and 10) appear instead to be scarcely influenced by rainfall.

In piezometer S3-VW, instead, the brief period oscillations caused by tides were not recognisable, while those caused by flood peaks were notably visible.

During the period monitored (18 September 2014–29 May 2015), the piezometric level appears to have dropped considerably. In particular, from an average of roughly 6.5 m during the period between October 2014–February 2015, this value moved to an average of 5 m between February 2015–October 2010, with a constantly decreasing trend. The negative trend corresponds with a diminution in overall rainfall recorded during the same period.

### 4.4. Numeric Hydrogeological Model

The results of the statistical calibration of the numeric model are summarised in Table 6 and Figure 11.

**Table 6.** Statistical parameters of the calibrated model.

| Statistical Parameter | Value |
| --- | --- |
| Number of head observations | 86 |
| Residual Mean | −0.23 |
| Residual Standard Deviation | 1.39 |
| Absolute Residual Mean | 1.06 |
| Residual Sum of Squares | 170 |
| RMS Error | 1.41 |
| Minimum Residual | −3.81 |
| Maximum Residual | 2.86 |

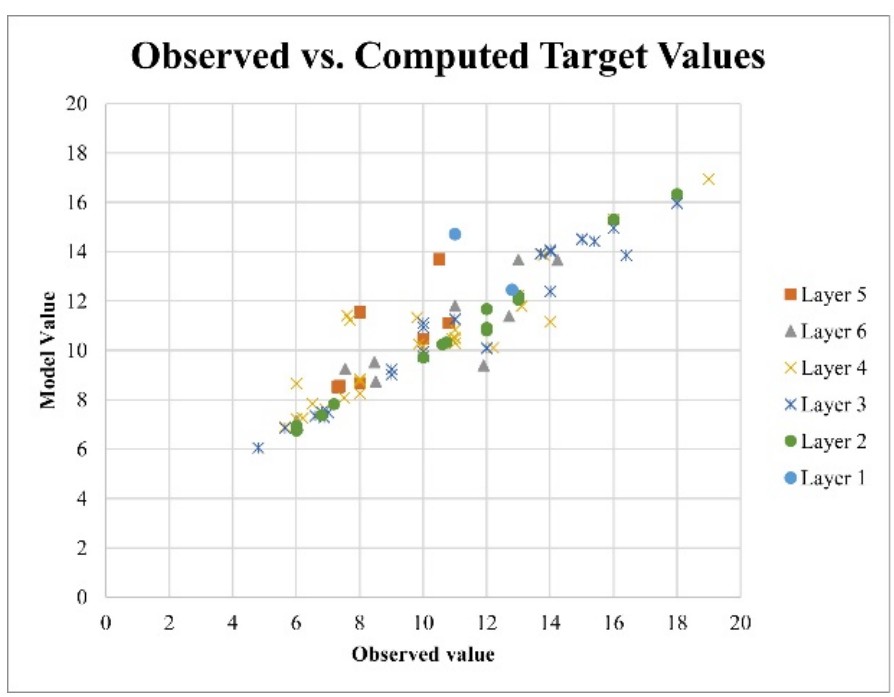

**Figure 11.** Scatterplot of observed versus computed target values.

The process of calibration produced an estimate of the optimum value of hydraulic conductivity in correspondence with the model targets; the estimated values of k were then interpolated, for each layer, by the software using Ordinary Kriging (Figure 12). In addition to the values of k, recharging was also calculated, arriving an optimum value of 0.00068 m/d.

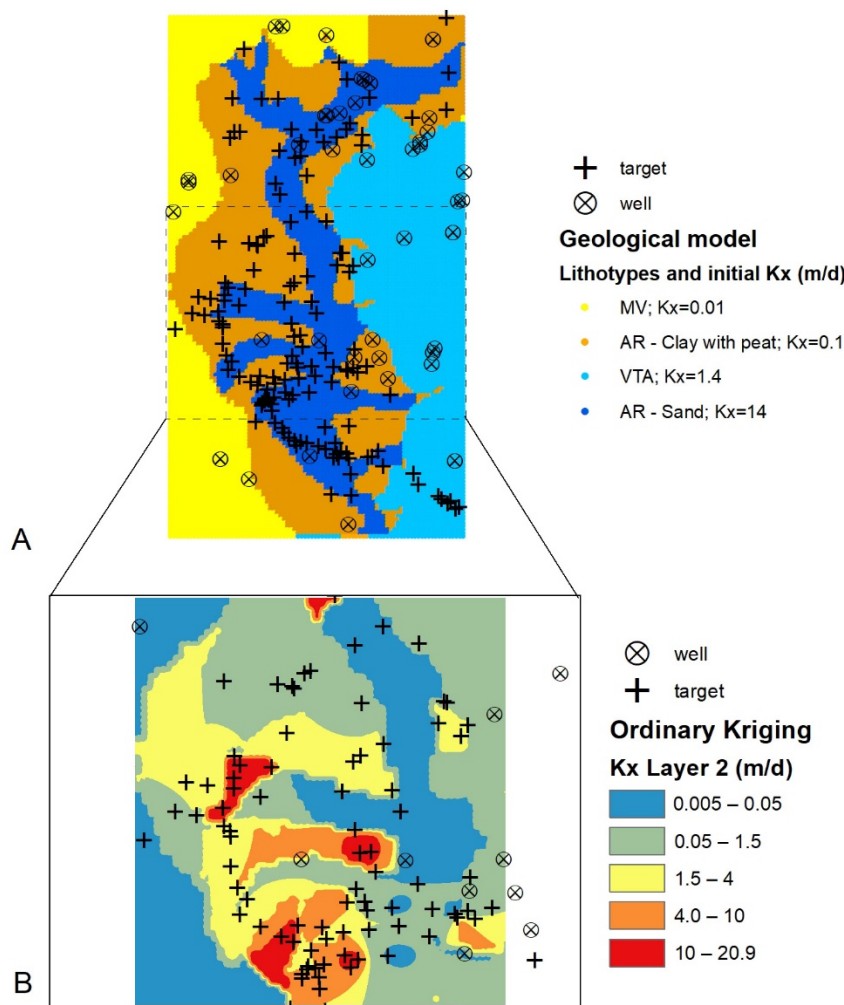

**Figure 12.** (**A**) Geological model from 5 to −5 m a.s.l.: colours represent the horizontal hydraulic conductivity (Kx) initially assigned to lithotypes. (**B**) Numerical groundwater model, Layer 2: colours represent the hydraulic conductivity values after sensitivity analysis and calibration.

## 5. Discussions

The results of the monitoring shed light on the diverse hydrogeological behaviour of the lithotypes as a function of the oscillations of the Tiber River. Despite being a considerable distance from the coastline (Table 2), the flow of the river is strongly influenced by tide oscillations; these oscillations can also be seen in the piezometric head level recorded in the gravely and sandy lithotypes. This data suggests the existence of a strong hydraulic connection between the aquifer hosted in both the base sands and gravels (piezometers S3-OS and S2-OS) and the river, in accordance with the geological structure reconstructed in the three-dimensional model of the lithotypes. The oscillations in the level of the clayey lithotype 4a (piezometer S3-VW) appear to follow the level of the river during flood events in a more dampened manner with respect to the other piezometers; what is more, no tide oscillations are recognisable; this suggests that lithotype 4a has a scarce connection with the river and a higher storage coefficient, with respect to sandy and gravely lithotypes, in accordance with the data in literature describing the behaviour of alluvial terrains. In particular, it has been observed that pressure waves such as tidal oscillations in unconfined aquifers tend to be damped because the storage coefficient is usually several orders of magnitude larger than the confined storage coefficient that governs pressure changes at depth ([52]). Moreover, if groundwater leakage occurs from a confined aquifer in vertical connection with an overlying unconfined aquifer, the tidal

amplitude of the groundwater head can be significantly reduced ([53]). In summary, the behaviour of groundwater oscillation suggests confining conditions for complexes 4b and 4d, and unconfined conditions for complex 4a; this is in agreement with the reconstruction in the 3D model. Moreover, a vertical leakage can be hypothesized from complex 4b–4d to 4a, which could be partially responsible for the tidal fluctuation damping. Finally, the oscillations in the level of piezometer S3-OS demonstrate that the exchange between the river and the water body of the base gravels is subject to inversions in direction based on the height of the fluvial stage: the gravely body can alternately receive water from the river or feed the river.

With respect to the influence of rainfall on the groundwater level oscillation, its effect is not visible in the short period peaks; however, the overall contribution of rainfall to the recharge of the aquifer is reflected in the trend of groundwater level. In particular, this trend was negative, this suggesting a decrease of the rainfall recharge to the aquifer.

The results of the calibration of the numeric model show a good correspondence between the head level observed and that simulated. All the same, the kriging mapping of the values of hydraulic conductivity resulting from the calibration of the model differs from the spatial reconstruction of the lithotypes based on the codification of the borehole lithofacies.

It must be mentioned that kriging estimates present a serious drawback, as the smoothing effect due to modelling parameters (e.g., the nugget effect and the variogram model selected), in which small values are usually overestimated and large values underestimated ([54]). Furthermore, the mapped heterogeneity distribution can result forced to achieve optimum parameter values. However, the spatial prediction for an observed location (target) is simply the observed value for that location, because kriging is an exact estimator. In the kriging map of Figure 12B, the calibrated k values in correspondence of targets are sometimes highly different from the initial k values: this is especially the case of some of the targets which were supposed to be located in the complex 4a (Figure 12A), with initial K = 0.1 m/d and calibrated K = 1.5–4 m/d. As a consequence, the map of Figure 12B seems to better match the spatial reconstruction of lithotypes.

This suggests that the geological model does not capture the real variability within the principal sedimentary facies, which needs to the object of further investigations. For example, the distribution of the lithotypes can be mapped beginning with the borehole data using stochastic statistic models. A greater accuracy in the spatial distribution of lithotypes (and therefore a greater accuracy of the conceptual model) may be of fundamental importance for converting the current numeric hydrogeological model into a transitory model. The simulation in a transitory regime would consent the reproduction of the movement over time of the piezometric levels, and also under conditions of flooding. Given the strict hydraulic relation between the complex of alluvial deposits and the Tiber River, it is nonetheless necessary to provide a robust geological model, to correctly simulate the response of the system to such stresses as flooding or intense rainfall. As a consequence, the stationary numeric model implemented in this study was useful for understanding how another conceptual model, based on a different distribution of lithotypes, could increase the accuracy of the simulation at the site specific scale and in a transient state.

In general, this study demonstrates the importance of structuring an iterative process for the construction of geological and hydrogeological models. In fact, the analysis of the lithofacies and corresponding lithotypes (phases 1 to 3, Figure 13) was used to create a 3D lithotype model, which, in turn, was described in hydrostratigraphic terms (phase 6) and was the subject of piezometric levels monitoring (phase 7). Then, the hydrostratigraphic model provided the base for a numerical groundwater model. Nonetheless, the hydrogeological model calibrated using data relative to the aquifer levels in a well suggested further methods for reconstructing sedimentary bodies that provide a better match with observed piezometric head levels (Figure 13). In this case, the numeric model in turn supported the refining of the geological model. A geological model opportunely reviewed based on the results of the numeric model may in turn provide the base for hydrogeological modelling in a transitory regime at the site specific scale. Finally, a detailed geological model can also

provide the base for other types of numeric models (e.g., differential settlement, subsidence, and saline intrusion). The iterative modelling process is thus very important when wishing to proceed with a detailed investigation of such aspects as geohazards, strictly dependent on the distribution of the lithotypes and hydrogeological behaviour.

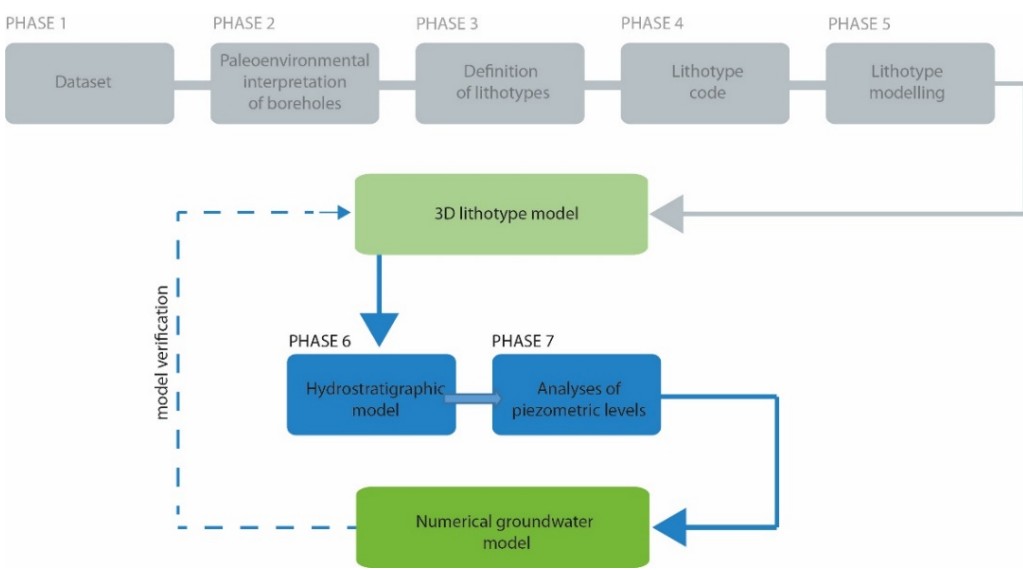

**Figure 13.** Flowchart showing the iterative process described in this study. The 3D lithotype model and the numerical groundwater model constitute the main outputs.

**Author Contributions:** Conceptualization, C.D.S., M.M. (Marco Mancini); methodology, C.D.S., M.M. (Marco Mancini), F.S.; Geological modelling software, F.S., M.M. (Marco Mancini).; Groundwater modelling software, C.D.S., M.D.; validation, C.D.S., F.S.; formal analysis, M.M. (Marco Mancini), C.D.S.; investigation, C.D.S., M.M. (Marco Mancini), M.M. (Massimiliano Moscatelli); resources, C.D.S., F.S., M.S., G.P.C.; data curation, F.S., C.D.S., G.P.C.; writing—original draft preparation, C.D.S., M.M. (Marco Mancini).; writing—review and editing, C.D.S., M.M. (Marco Mancini), F.S., M.M. (Massimiliano Moscatelli), M.S.; visualization, F.S., C.D.S., M.M. (Marco Mancini); supervision, G.P.C.; project administration, F.S.; funding acquisition, F.S. All authors have read and agreed to the published version of the manuscript.

**Funding:** This research was funded by Regione Lazio, FILAS project F87112000080007 "TIBER–Innovazione nel campo geotecnico per la definizione di strumenti, metodologie operative e procedure finalizzate alla realizzazione di un nuovo modello di sottosuolo (modello integrato)" project leader Francesco Stigliano.

**Data Availability Statement:** Not applicable.

**Acknowledgments:** The authors would like to thank Francesco Versino and Stefano Mastrototaro for their precious technical support during the field surveys. The 3D model and the cartographic elaborations were developed by the GIS Lab (CNR-IGAG; https://www.igag.cnr.it/lista-laboratori/labgis/, last accessed 1 December 2021); the hydrogeological elaborations were developed by the Laboratorio di idrogeologia quantitativa e modellazione numerica (CNR-IGAG; https://www.igag.cnr.it/lista-laboratori/laboratorio-di-idrogeologia-quantitativa-e-modellazione-numerica/, last accessed 1 December 2021). Moreover, authors are grateful to reviewers, whose suggestions allowed significant improvement to the manuscript.

**Conflicts of Interest:** The authors declare no conflict of interest.

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
