# Peer review of "From Lithological Modelling to Groundwater Modelling: A Case Study in the Tiber River Alluvial Valley"

_geosciences, doi:10.3390/geosciences11120507_

Round 1

Reviewer 1 Report

This paper discusses groundwater modeling using a very robust sediment analysis and a complete model. The contributions of this model are clear. The only point that can be improved in my opinion would be the form of some parts, such as the introduction. Furthermore, a mix is unclear in my opinion between the geological setting and the new results.

However, the article is globally clear and deserves only minor corrections in my opinion.

1 . Introduction

The structure of the introduction deals mainly with the administrative framework of the project, a more precise introduction on the real interests and stakes of the study would be appreciated.

  1. Setting

This part is a mixture between settings and, probably, new results as suggested by the sentence “three new boreholes” line 100 . If there is any new result, it should be presented in results section.

Line 62 : please cite the figure 1.

Lines 74-78 : Please rephrase, the sentence is not clear and it contains a dot, line 77, that should not appear.

Line 79: Figures 3a,b and c should be cited here as representation of the TDS.

Line 84: What do you mean by urban section? Please locate it on the map.

Line  100 and below: “three new boreholes”, interpretations should take part of results section and method should be detailed in material and method section is those results are new. If not, sources should be cited here as in the figure 2 legend.

Lines 100 - 154 : This part contain a lot of description. Here, good illustrations/light explanation based on the figure 3b should increase the reader's understanding.

Line 160 : section 2.2 should be integrated in both Mat and Meth and results sections.

  1. Mat & Methods

Line 316 : Please indicate the aim in establishing this lithotype. Moreover, please add the verification of the conceptual model on the workflow of the Figure 5.

The method used for lithotype is clearly described and seems complete.

Line 417: The aim of the model is presented here, it should be explained before.

Line 430: “1m/day”, you have to be more precise: what study allows you to choose this number, or what measure?

Reviewer 2 Report

Review "From Lithological Modelling to Groundwater Modelling: A Case Study in the Tiber River Alluvial Valley" by Di Salvo et al.

General comments

Introduction is an extended abstract. It does not present the work in the international context, nor discusses novelty of the work or how it advances knowledge or how it relates to international literature.

Section 2.1 is so rich in details to the level that is hard to follow at some points. It reads more as M&M rather than intro material, in particular section  2.2, which described the new boreholes drilled (S, S2, S3).

Figure 2. should be part of your research results/interpretation of drilling logs.

Major comments

Section 4.2 presents interesting results. However, discussion lacks depth. The dampening of the amplitude for head oscillations in unit 4a needs some explanation, could it be other units impacting vertical connectivity? overlying units 4b or 4d?

L519-520 discussed that changes in heads in S3 are not influenced by rainfall. Then, L526-527 suggest decreasing trend in heads corresponds (correlates?) with decreases in rainfall for thee same period. This needs further explanation/justification as it is a contradiction in the argument.

L578-583, this raises the issue of post-calibration kriging mapping as validation of variability/heterogeneity in sedimentary facies. Authors need to acknowledge that kriging interpolation will either smooth the variability depending on modelling parameters (e.g. nugget effect, and variogram model selected) or will force heterogeneity (e.g. bulls-eye patterns) to achieve optimum parameter values in a mathematical sense. So, it might well be that the sedimentary model is correct and the kriging interpolation is  flawed.

Minor comments

Avoid acronyms in abstract

L23-24. "..numerical hydrological level in a stationary regime." This probably needs rephrasing as it is unclear.

L35-37. English

L42, groundwater

L85, into

L90, 93, 99, correlation

L100, depth of boreholes?

L155. Figure 1. Features "7" and "8" need to be identified

L214. Figure 3c. Borehole stratigraphy is missing in cross section. It would be interesting to include water table/piezometric surfaces in the cross sections as well

L253. Figure 4. Please describe hydrogeological complexes 1 to 5 in legend

L406. "screened or slotted interval" might be more precise

L411. for how long?

L426-427. Something is missing?

L440 The whole section 4.1 does not discuss nor present results and should be moved to M&M section. What was the main result of the geolithological model? is it Fig 8? if so, discuss that in terms of literature (e.g. palaeoenvironments?, validation of previous lithological models? relevance for hydrogeological modelling? etc.)

L523, define monitoring period

L550. Figure 11. Make axes symmetrical to facilitate interpretation

L551. Figure 12. Please check negative values for Kx Layer 2

L582-583, something missing

L587, transitory? did you mean transient?

Recommendation

Overall this is an interesting piece of work. It presents and applies a clear methodological framework. Discussion is supported by the data/results presented. My only concern is that it lacks the structure of a scientific article and sometimes it reads as a summary of a technical report. For example, this work is not presented in the context of international literature, it did not discuss the novelty of the work nor how it advanced the knowledge. I would suggest to the authors to expand the literature review to give context to the work, explicitly state the novelty and contribution of this research. Then, clearly distinguish between material&methods and results. Based on the above review I would suggest the article for publication after these comments are addressed.

Reviewer 3 Report

The manuscript “From Lithological Modelling to Groundwater Modelling: A Case Study in the Tiber River Alluvial Valley” mainly describes the establishment of geological and hydrogeological models of Tiber River and the relationship between them. The study has positive significance for Tiber River basin and model construction. The structure of the manuscript is relatively complete, and there are no obvious problems with the use of English., the research idea is basically clear and the scheme design is basically reasonable, but there are several points that needed to be clearly explained, which are as follows:

  1. Without highlighting and explaining the innovation of the article.
  2. An explanation of legends 7 and 8 is missing in Figure 1.
  3. Add a scale in Figure 3 and adjust the layout of Figure 3C.
  4. Where is the heading 3.3.
  5. Add the title number of line 342 and so on.
  6. Modify font confusion in line 477 and so on.
  7. Carefully check the citation format of the references.
